# A critique of the hypothesis that CA repeats are primary targets of neuronal MeCP2

Kashyap Chhatbar[1,2,*], John Connelly[1,*], Shaun Webb[1], Skirmantas Kriaucionis[3], Adrian Bird[1]

**The DNA-binding protein MeCP2 is reported to bind methylated cytosine in CG and CA motifs in genomic DNA, but it was recently proposed that arrays of tandemly repeated CA containing either methylated or hydroxymethylated cytosine are the primary targets for MeCP2 binding and function. Here we investigated the predictions of this hypothesis using a range of published datasets. We failed to detect enrichment of cytosine modification at genomic CA repeat arrays in mouse brain regions and found no evidence for preferential MeCP2 binding at CA repeats. Moreover, we did not observe a correlation between the CA repeat density near genes and their degree of transcriptional deregulation when MeCP2 was absent. Our results do not provide support for the hypothesis that CA repeats are key mediators of MeCP2 function. Instead, we found that CA repeats are subject to CAC methylation to a degree that is typical of the surrounding genome and contribute modestly to MeCP2-mediated modulation of gene expression in accordance with their content of this canonical target motif.**

## Introduction

MeCP2 is a chromatin protein that is abundant in neurons and essential for brain function. Initially identified through its affinity for 5-methylcytosine in DNA, the exact nature of its DNA targets has periodically been debated and challenged. Based on in vivo and in vitro data from several laboratories, targeting of MeCP2 to DNA depends on the presence of mC in two motifs: mCG and mCA (the latter mostly in the trinucleotide mCAC) (Lewis et al, 1992; Ayata, 2013; Guo et al, 2014; Gabel et al, 2015; Lagger et al, 2017; Cholewa-Waclaw et al, 2019; Connelly et al, 2020). Independent studies suggest that the presence of mCA in neurons is essential, in part at least because of its affinity for MeCP2 (Gabel et al, 2015; Lavery et al, 2020; Tillotson et al, 2021). For example, mice expressing a modified form of MeCP2 that does not interact with mCA but can still bind mCG develop severe Rett syndrome–like phenotypes (Tillotson et al, 2021). In vitro, MeCP2 also binds to the hydroxymethylated (hm) motif hmCAC (Lagger et al, 2017), but the significance of this interaction has been unclear because of the apparent rarity of this modified trinucleotide in brain genomes (Lister et al, 2013).

This scenario is challenged by a recent proposal that arrayed tandem repeats of the dinucleotide CA are critical MeCP2 targets, exceeding in importance both mCG and isolated mCA moieties as mediators of MeCP2 function (Ibrahim et al, 2021). As $[CA]_n$ repeat blocks are relatively frequent, it is argued that their proximity to genes provides high-affinity "landing pads" through which MeCP2-dependent gene regulation is mediated. Mechanistically, it is suggested that MeCP2 binding to occasional mCA or hmCA moieties within $[CA]_n$ repeats can seed cooperative MeCP2 binding across the entire array, including non-methylated CA motifs. Here, we further investigate the relationship between MeCP2 and CA repeats. Our findings do not offer support for the claim that cytosine modification is enriched at $[CA]_n$ arrays or that MeCP2 is preferentially bound at these repeat blocks in brain cell nuclei. Moreover, we find that the effects of MeCP2 deficiency on transcription in various brain regions do not correlate with proximity to $[CA]_n$ repeats but instead strongly correlate with local mCAC frequency.

## Results

### Absence of enrichment of modified cytosine in CA repeats

The mouse genome (version mm9) contains ~320,000 $[CA]_n$ arrays (minimum 10 base pairs) of variable length with an average of 25 CA repeats each. It has been reported that CA is more frequently methylated or hydroxymethylated within $[CA]_n$ repeat arrays than elsewhere in the genome (Ibrahim et al, 2021), but this comparison did not take account of the preferential methylation of CAC trinucleotides in neurons (Lagger et al, 2017). Although CAC is necessarily very abundant within $[CA]_n$ repeats, isolated CA motifs elsewhere in the genome may or may not have C in the third position. Recognizing that the trinucleotide sequence CAC is the preferred target of non-CG methylation in neurons and is also a

[1]Wellcome Centre for Cell Biology, University of Edinburgh, The Michael Swann Building, Edinburgh, UK   [2]Informatics Forum, School of Informatics, University of Edinburgh, Edinburgh, UK   [3]Ludwig Institute for Cancer Research, University of Oxford, Oxford, UK

Correspondence: a.bird@ed.ac.uk
*Kashyap Chhatbar and John Connelly contributed equally to this work.

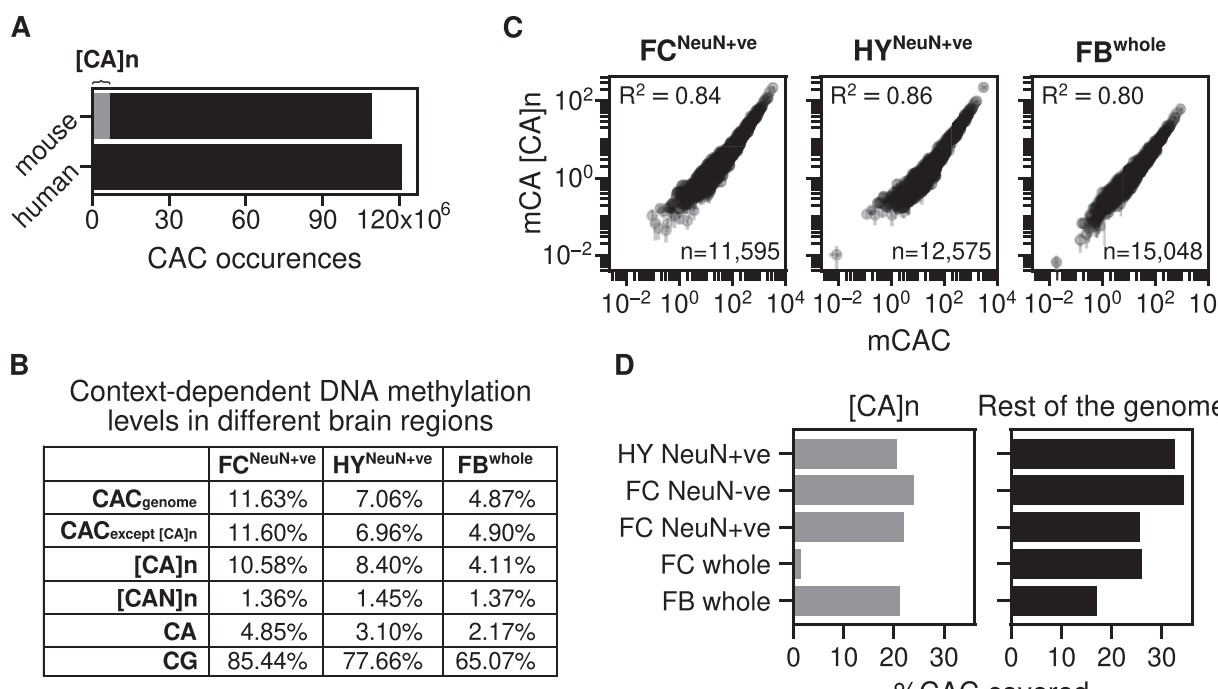

**Figure 1. Absence of enrichment of modified cytosines in CA repeat arrays.**
**(A)** Number of CAC occurrences across the mouse and human genome. The grey area corresponds to the number of CAC occurrences within CA repeat arrays.
**(B)** Genome-wide average DNA methylation levels for different cytosine contexts: CAC across the whole genome; CAC across the whole gene except [CA]$_n$; [CA]$_n$ dinucleotide repeats; [CAN]n trinucleotide repeats (where N is A, T, or G); CA; and CG. DNA methylation levels were quantified from three mouse brain regions: sorted NeuN+ve nuclei from the hypothalamus (HY) (Lagger et al, 2017); sorted NeuN+ve nuclei from the frontal cortex (FC) (Lister et al, 2013) and forebrain (FB) (Boxer et al, 2020). **(C)** mCAC number per gene plotted against mCA number per gene exclusively in CA repeats and Pearson correlation ($R^2$) is calculated for three mouse brain regions: sorted NeuN+ve nuclei from the hypothalamus (HY) (Lagger et al, 2017); sorted NeuN+ve nuclei from the frontal cortex (FC) (Lister et al, 2013) and forebrain (FB) (Boxer et al, 2020). **(D)** Percentage of CAC sites with adequate bisulfite sequence coverage within CA repeat loci and across the mouse genome. The coverage threshold for sorted NeuN+ve hypothalamus (HY) (Lagger et al, 2017) and whole forebrain (FB) (Boxer et al, 2020) is at least five reads, and threshold for sorted NeuN+ve, NeuN−ve, and whole frontal cortex (FC) (Lister et al, 2013) is at least 10 reads.

target for MeCP2 binding (Lagger et al, 2017; de Mendoza et al, 2021), we determine that ~6% of all CAC motifs in mouse are found within [CA]$_n$ repeats (Fig 1A). Interestingly, humans (version chm13-v1.1) possess a much lower proportion of [CA]$_n$ repeats: ~55,000 [CA]$_n$ arrays amounting to only ~1% of all CAC motifs in the genome (Fig 1A). Using published data for three mouse brain regions, we confirmed that mCA occurs at a lower frequency outside [CA]$_n$, but the frequency of mCAC for each brain region was similar within and outside the repeat arrays (Fig 1B). To determine whether the levels of cytosine modification in [CA]$_n$ arrays match the level of mCAC nearby, we plotted mCAC number per gene against mCA number exclusively within CA repeats (Fig 1C). The results showed a strong correlation, indicating that the local density of mCAC in genes is similar regardless of whether the trinucleotide is isolated or within a CA repeat array. We conclude from these findings that, although [CA]$_n$ arrays are subject to CAC methylation, they are not targeted preferentially compared with the surrounding genome but tend to adopt a level of mCAC that reflects the neighbouring DNA.

This conclusion assumes that the level of cytosine modification in [CA]$_n$ arrays has not been systematically underestimated by bisulfite sequencing because of reduced coverage of repetitive sequences. To test for under-representation in published bisulfite sequence data, we compared the fraction of CAC covered in [CA]$_n$ repeats versus the rest of the genome. The results show little difference for forebrain and

reduced coverage of [CA]$_n$ arrays (<2-fold) in NeuN+ve frontal cortex and hypothalamus (Fig 1D). Slightly lower coverage of CA repeats has little effect on estimates of DNA methylation levels as, even when inadequately covered sequences are excluded, the number of CACs that are reliably detected (1,913,430) is more than sufficient to allow accurate determination of their modification status. The exception to this was the whole frontal cortex dataset (280,015 CACs reliably detected), where bisulfite sequence data gave much lower coverage of [CA]$_n$ repeats (Lister et al, 2013). Importantly, only this sparsely covered dataset was analyzed by Ibrahim et al (2021). High bisulfite sequence coverage was obtained with purified cortical NeuN+ve (neuronal) and NeuN−ve (mostly non-neuronal) nuclei from the same study (Lister et al, 2013) (Fig 1D), demonstrating that CA repeats are not intrinsically under-represented in the cortex by this technology. Leaving aside the outlier dataset from whole frontal cortex, the evidence indicates a modest bias by bisulfite sequencing against [CA]$_n$ repeats. Despite this effect, CA repeat coverage is sufficient to strongly support the conclusion that methylation of mCAC is similar between CA repeats and the rest of the genome.

## Absence of enrichment of MeCP2 binding at CA repeats in the brain

We next asked whether MeCP2 is preferentially associated with CA repeat arrays (Ibrahim et al, 2021). MeCP2 ChIP of mouse brain

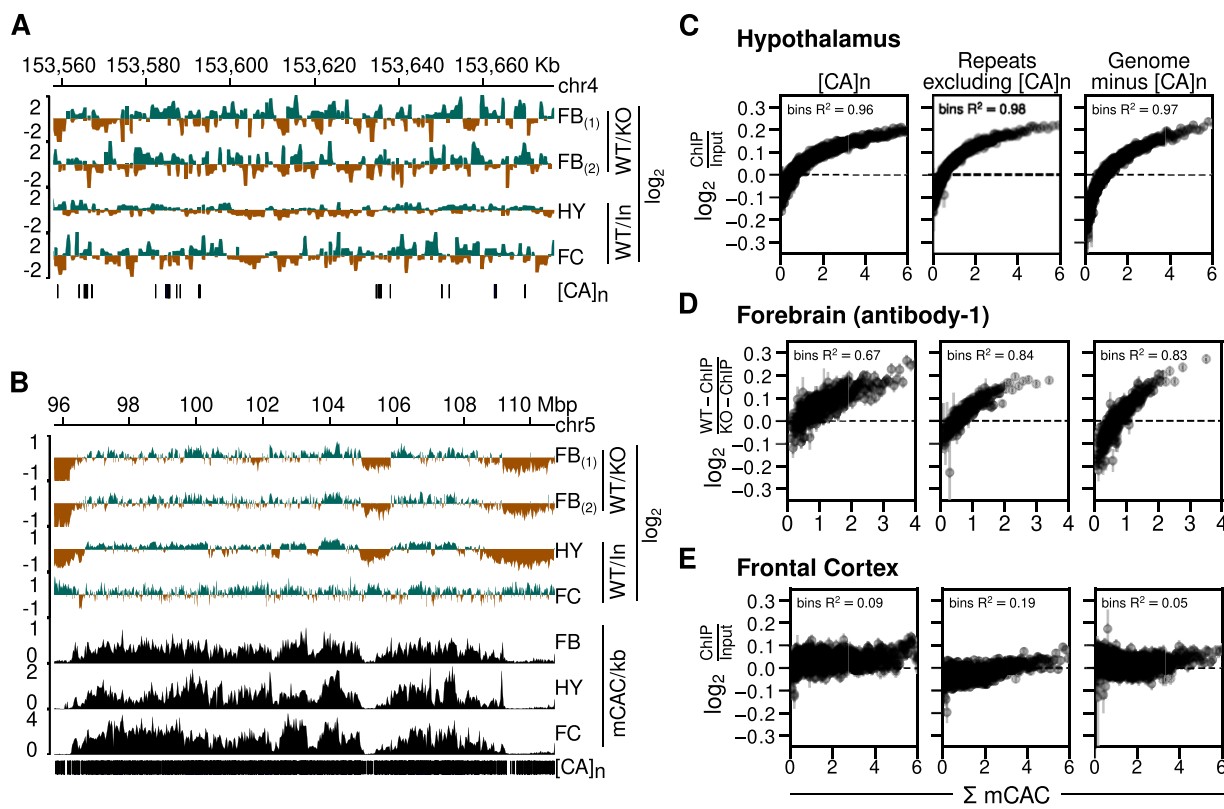

**Figure 2. Absence of enhanced MeCP2 binding at CA repeat arrays in brain.**
**(A)** Genome browser screenshots of mouse chr4:153558029-153676791 (version mm9) showing *Mecp2 wild-type* (*WT*) ChIP signal normalized to *Mecp2 null* (*KO*) ChIP signal for the forebrain (FB) using two distinct antibodies (Boxer et al, 2020), *Mecp2 WT* ChIP signal normalized to input chromatin for the hypothalamus (HY) (Chen et al, 2015) and frontal cortex (FC) (Gabel et al, 2015). Vertical strokes (bottom row) show location of CA repeat arrays. **(B)** As described for panel (A) but using coordinates at chr5: 95751179-110732516. In addition, DNA methylation tracks showing mCAC/kb for sorted NeuN+ve nuclei from the forebrain (FB) (Lister et al, 2013), hypothalamus (HY) (Lagger et al, 2017), and sorted NeuN+ve nuclei from the frontal cortex (FC) (Lister et al, 2013). **(C)** MeCP2 ChIP signal normalized to input chromatin in mouse hypothalamus (Chen et al, 2015) plotted against bins of increasing levels of DNA methylation at CAC (Lagger et al, 2017) (shown as mean ± standard error of mean) within three different genomic sequence categories. Panels from left to right represent CA repeat loci, simple repeat loci other than CA repeats as identified by RepeatMasker, and 500,000 randomly chosen 1 kb genomic windows which do not overlap with CA repeats. $R^2$ values indicate squared Spearman correlation from binned mean values of MeCP2 ChIP enrichment and binned mean values of mCAC. **(D)** As described for panel (C) but using data for mouse forebrain (Boxer et al, 2020). MeCP2 ChIP signal is normalized to ChIP in *Mecp2 KO*. **(E)** As described for panel (C) but using data for mouse frontal cortex (Gabel et al, 2015) and Lister et al (2013).

reproducibly reveals relatively uniform genome occupancy with few prominent peaks (Chen et al, 2015; Gabel et al, 2015; Lagger et al, 2017). This has been interpreted to reflect the high frequency throughout the neuronal genome of short MeCP2 target sites, mCG and mCAC (Lagger et al, 2017). In contrast, Ibrahim et al report that MeCP2 ChIP-Seq reads in cultured MEFs are concentrated in prominent peaks coincident with CA repeat clusters (Ibrahim et al, 2021). Given the importance of MeCP2 function and the exceptionally high abundance of hmC and mC in neurons, equivalent peaks at [CA]n might be expected in the brain. However, published ChIP data do not support this prediction. As an example, the MeCP2-binding profile (normalized to the *KO* or input ChIP profiles) across the same ~120 kb region of the mouse genome that was illustrated for MEFs (Ibrahim et al, 2021) failed to highlight [CA]n arrays (Fig 2A). In view of uncertainties regarding the initial MEF data (see the Discussion section), our findings question the evidence for preferential localization of MeCP2 to CA repeats.

We also visualized enrichment of MeCP2 at lower resolution across a much larger (15 megabase) region of mouse chromosome 5 (Fig 2B). The results agree with previous reports of a fluctuating

distribution of MeCP2 across the genome that broadly tracks the density of mCAC (Lagger et al, 2017). Although [CA]n repeat arrays are not apparent as prominent sites of MeCP2 binding, their high density at this resolution makes it difficult to discern by inspection alone whether they are preferred. In addition, the global distribution of bound MeCP2 across the neuronal genome limits the value of traditional peak analysis methods to define the binding pattern (Lagger et al, 2017). To reveal the relationship of ChIP signal to CA repeats versus other genomic regions, we plotted bins of $\log_2$ fold-change in MeCP2 binding (normalized to *KO* or input) versus mCAC frequency for three DNA sequence categories: (i) [CA]n; (ii) repeated sequences excluding [CA]n; and (iii) the rest of genome excluding [CA]n. We drew upon published datasets derived from brain regions for which matching ChIP and bisulfite data were available and again normalized to *KO* ChIP signal or input (Lister et al, 2013; Chen et al, 2015; Lagger et al, 2017; Boxer et al, 2020). The results showed that for the hypothalamus, MeCP2 occupancy clearly rose as mCAC frequency increased (Fig 2C). If [CA]n arrays were preferential targets for MeCP2 binding, we would expect them to show elevated ChIP enrichment compared with the other

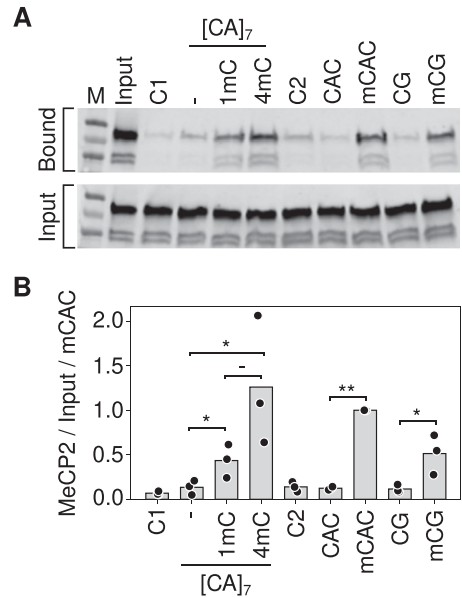

**Figure 3. MeCP2 binding to [CA]₇ is dependent on cytosine methylation.**
**(A)** Example of a pulldown assay for MeCP2 binding using biotin-tagged double-stranded DNA oligonucleotides incubated with mouse brain nuclear extracts (see the Materials and Methods section). Unrelated probes C1 and C2 contained no mC. Probe CAC contained 10 non-methylated CACs on one strand, all methylated in mCAC. Alternative versions of the [CA]₇ probe contained 0, 1, and 4 mCAC motifs labelled as, 1 and 4 mC respectively. **(B)** Quantification of the triplicate data exemplified in (A). Significance was estimated using a paired *t* test (*Pval < 0.05, **Pval < 0.01).

genome categories, but the three DNA sequence types showed closely similar profiles in each brain region. The same conclusion can be drawn from data from the forebrain, which is represented by two high-coverage datasets using independent anti-MeCP2 antibodies (Fig 2D). A fourth dataset derived from mouse frontal cortex (Lister et al, 2013; Gabel et al, 2015) also showed a trio of near-identical ChIP plots, but in this case, the relationship to mCAC frequency was less striking (Fig 2E). Rank correlations derived from the averaged bins across the whole genome excluding CA repeats gave $R^2$ values of 0.97 for the hypothalamus and 0.83 for the forebrain, indicating strong dependence between MeCP2 enrichment and mCAC. In the case of frontal cortex, an $R^2$ value of 0.05 revealed only modest enrichment for highly methylated regions of the genome. Using peak enrichment to indicate the relative coverage of ChIP-Seq datasets, we could identify 271,334 and 236,330 MeCP2-enriched regions in hypothalamus (Chen et al, 2015) and forebrain (Boxer et al, 2020) ChIP datasets, respectively, whereas the frontal cortex data (Gabel et al, 2015) detected only 37,817 enriched regions. This ~7-fold decrease suggests lower resolution of the frontal cortex ChIP data which may contribute to its different profile (Fig 2B) and modest correlation between MeCP2 binding and DNA methylation (Fig 2E). Regardless of differences, all three datasets failed to reveal evidence of enhanced binding of MeCP2 at [CA]ₙ, as the intensity of ChIP signal at [CA]ₙ was approximately equivalent to that in other parts of the genome.

Based on electrophoretic mobility shift assays, it was further proposed that cooperative binding across [CA]ₙ arrays is facilitated by the affinity of MeCP2 for non-methylated [CA]ₙ (Ibrahim et al, 2021). This recalls early reports (Weitzel et al, 1997) that certain

truncated variants of MeCP2 can bind to CA/TG-rich probes in vitro, although recent evidence failed to validate this mode of binding with full-length MeCP2 either in vitro or in vivo (Connelly et al, 2020). Ibrahim et al (2021) reported MeCP2 binding to longer non-methylated CA repeat tracts specifically [CA]₇. Using a pulldown assay for native MeCP2 binding in mouse brain extracts, however, we failed to detect enhanced MeCP2 binding to [CA]₇ compared with control DNA probes that lacked CA repeats or in which CA was part of a CAGA repeat array (Fig 3A and B). Introduction of one mC residue into the [CA]₇ tract significantly increased MeCP2 binding. Moreover, binding displayed a trend towards further enhancement when three more cytosines in the array were methylated. This result is compatible with a linear rather than a cooperative relationship between the amount of mCAC and MeCP2 binding. We noted that 10 mCACs in a non-repetitive probe did not appear to further enhance binding compared with four mCACs. Potential explanations for this apparent plateau include steric interference of closely proximal mCAC sites, probe length, etc. Unfortunately, the variability between experiments prevented us from exploring these alternatives quantitatively. Overall, our results fail to confirm an intrinsic affinity of MeCP2 for non-methylated [CA]₇ in vitro, and they suggest that addition of one mCAC motif is not sufficient to cause cooperative MeCP2 binding across a [CA]₇ array, as further methylation of the [CA]₇ probe further enhances binding.

### Minor effect of CA repeats on MeCP2-mediated gene regulation

We next tested the relationship between gene expression changes in the MeCP2-deficient brain and the frequency of [CA]ₙ repeat clusters within gene bodies, drawing on data from independent studies of mouse hypothalamus, forebrain, and cortex (Chen et al, 2015; Gabel et al, 2015; Boxer et al, 2020). To investigate the effect of enriched [CA]ₙ repeat clusters on transcription, we asked whether significantly up- or down-regulated gene bodies were enriched for [CA]ₙ. Box plots of these gene categories failed to show obvious relationship between % [CA]ₙ in up- or down-regulated genes compared with random non-regulated gene bodies (Fig S1A). We also plotted the percentage of all transcription unit nucleotides that belong to [CA]ₙ arrays against the fold change in gene expression when MeCP2 is absent. This differs from a previous analysis (Ibrahim et al, 2021) by taking into account the direction of transcriptional change and testing multiple brain datasets. Again, the results showed no obvious correlation between the differing levels of [CA]ₙ in the gene body and changes in gene expression in these brain regions (Fig 4, left panels). In contrast, we found that the number of mCAC motifs per gene, either including or excluding [CA]ₙ repeat clusters, correlates positively with the average magnitude of gene up-regulation in the mutant brain (Fig 4, middle panels), supporting the notion that MeCP2 binding to this methylated motif restrains gene expression and confirming previous findings (Kinde et al, 2016; Lagger et al, 2017). Although gene length correlates with gene misregulation in *Mecp2 KO* (Gabel et al, 2015), a positive correlation with mCAC persisted when mCAC motifs per gene were normalized to gene length (Fig S1B). This suggests that gene length does not sufficiently explain the positive correlation with mCAC in the absence of MeCP2. The strong relationship influenced by mCAC motifs, which is unaffected by inclusion or exclusion of [CA]ₙ arrays, is not expected if CA repeats were the primary drivers of MeCP2-mediated gene

**A**

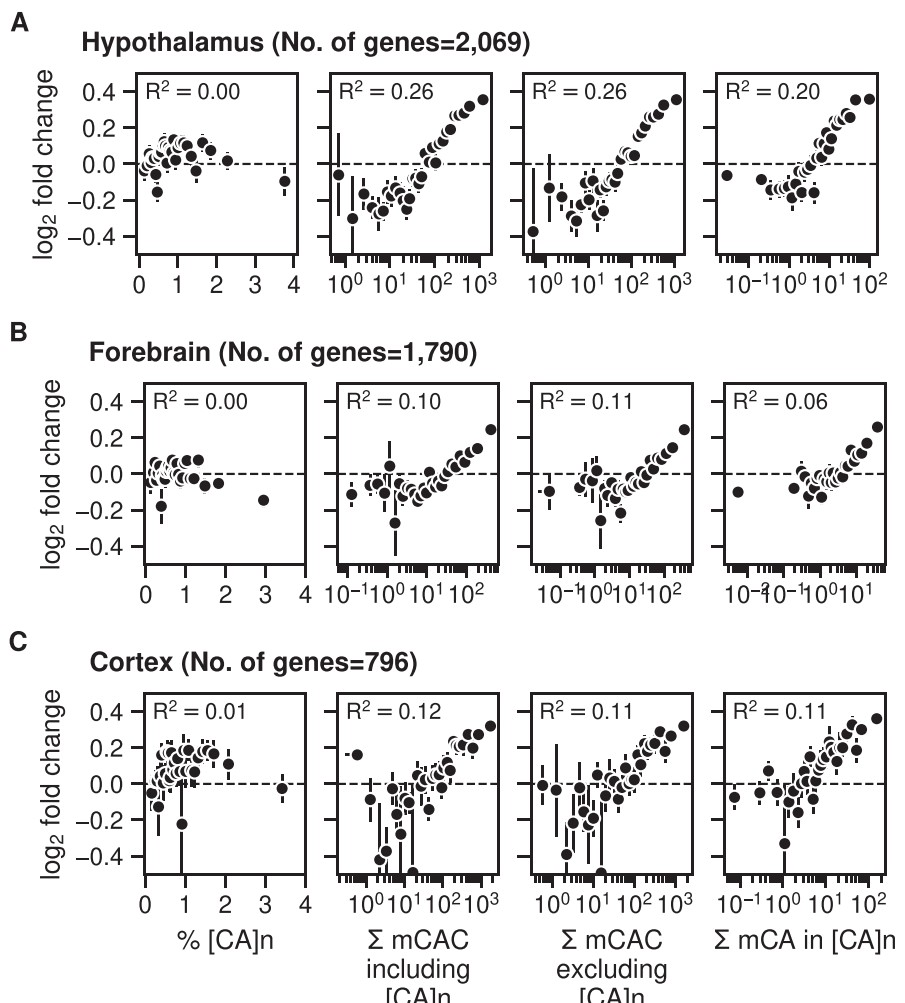

**Figure 4. Gene expression increases in *Mecp2 KO* brain regions according to levels of mCAC but does not correlate with the presence of CA repeat arrays.** **(A)** Mean $\log_2$ fold-change in gene expression in *Mecp2 KO* versus MeCP2 *WT* in mouse hypothalamus (differentially regulated genes were defined by *P*adj < 0.05) (Chen et al, 2015) plotted against the percentage of CA repeats within the gene body (left panels); bins of increasing mCAC number per gene including or excluding CA repeats (Lagger et al, 2017) (centre panels); and bins of increasing mCA per gene exclusively within CA repeats (Lagger et al, 2017) (right panels). Spearman rank correlation ($R^2$) is calculated using unbinned values for every panel. **(B)** As described for panel (A) above but using data for mouse forebrain (differentially regulated genes were defined by *P*adj < 0.05) (Boxer et al, 2020). **(C)** As described for panel (A) above but using data for mouse frontal cortex (differentially regulated genes were defined by *P*-value < 0.05) (Lister et al, 2013; Gabel et al, 2015).

regulation. Because CA repeats are subject to DNA methylation at the same level as dispersed CAC motifs (see Fig 1), we expected that the presence mCAC within $[CA]_n$ tracts would correlate with gene expression. This was confirmed when the number of mCA motifs in $[CA]_n$ repeat blocks was plotted against the fold change in transcription between *Mecp2 KO* and *WT* brain regions (Fig 4, right panels). To quantify these findings, we calculated the rank correlation with $\log_2$ fold-change in gene expression for unbinned values. $R^2$ values were consistently higher in plots of total mCAC number per gene (including or excluding $[CA]_n$) than for % $[CA]_n$ (Fig 4, middle panels). When the level of CA methylation in $[CA]_n$ was taken into account (Fig 4, right panels), the correlation with transcriptional change did not exceed that of mCAC elsewhere in the genome. Our findings do not support the hypothesis that cytosine modification in $[CA]_n$ has a heightened impact on gene regulation.

## Discussion

We investigated the possibility that CA repeats are preferred targets of MeCP2 binding because of enrichment of mC or hmC, potentially

leading to cooperative binding across the entire tract. In the mouse brain, we detected neither enrichment of CAC modification nor obvious accumulation of bound MeCP2 within $[CA]_n$ repeat clusters. Instead, the data derived from several independent studies indicate that although arrays of $[CA]_n$ do acquire cytosine methylation, the average level of mCAC within them is typical of the surrounding genome. In agreement with this finding, the level of MeCP2 binding within $[CA]_n$ repeats is as expected given the frequency of its target motif mCAC. In the absence of MeCP2, gene expression is up-regulated according to mCAC density as reported previously (Guo et al, 2014; Chen et al, 2015; Kinde et al, 2016; Lagger et al, 2017; Cholewa-Waclaw et al, 2019; Boxer et al, 2020), but we found no obvious correlation with the proportion of gene bodies made up of $[CA]_n$ repeat blocks unless the frequency of mCAC was taken into account. We conclude that the effect of $[CA]_n$ tracts on gene expression depends on the amount of mCAC that they contain.

Several further considerations lead us to question the proposed link between hmC and MeCP2 (Ibrahim et al, 2021). A major reservation concerns the use of antibody detection of hmC, which provided the initial stimulus for the hypothesis of Ibrahim and colleagues (Ibrahim et al, 2021). In their experiments,

immunoprecipitation with an antibody directed against hmC revealed prominent apparent peaks of hmC coincident with CA repeat clusters in MEFs. This result is unexpected, as levels of hmC are usually very low in dividing cultured cells compared with neurons. More importantly, others have reported "serious flaws" in the MeDIP method that lead to erroneous reporting of hmC even when both mC and hmC are known to be absent (Lentini et al, 2018). Strikingly, the resulting false positives, which account for 50–99% of regions identified as "enriched" for DNA modifications, are predominantly found at unmodified short repeat arrays, in particular [CA]$_n$. In view of this potentially serious caveat, the evidence for hmC at CA repeats in this MEF cell line must be considered provisional, pending independent biochemical validation.

A second concern relates to the biochemical evidence for the binding specificity of MeCP2. In support of the hypothesis that hmC or mC in CA repeat arrays are the primary targets of MeCP2, Ibrahim et al report that [hmCA]n repeats have a seven-fold higher affinity for MeCP2 in vitro than for the canonical MeCP2 target motif mCG (Ibrahim et al, 2021). However, rather than using symmetrically methylated mCG/mCG, which is a validated target sequence, as a comparator, the authors chose hemi-methylated mCG/CG. The affinity of MeCP2 for hemi-methylated mCG/CG is reproducibly little more than background (Valinluck et al, 2004; Hashimoto et al, 2012; Lei et al, 2019), making this an invalid control. Estimated dissociation constants for the interaction between MeCP2 and symmetrical mCG/mCG are somewhat variable in the literature depending on the details of the assay, ranging from a low of 400 nM (Lei et al, 2019) to 15 nM (Valinluck et al, 2004) or 10 nM (Hashimoto et al, 2012). Notably, these published affinities for mCG/mCG are similar to or higher than the affinity for hmCA-containing repeats reported by Ibrahim et al (2021) (410 nM).

Finally, the extreme rarity of hmCA in the brain is difficult to reconcile with its hypothetical pivotal role. A limitation of many brain methylome datasets is that only bisulfite sequence analysis was performed and therefore mC and hmC are not distinguished. It is clear, however, that although the abundance of neuronal mCA is similar to that of mCG in the brain, the vast majority of neuronal hmC is confined to hmCG (Lister et al, 2013). For example, using a non-destructive method for hmC detection, it was shown that 97.5% of hmC in excitatory neurons is in hmCG, with less than 2.5% in hmCA (Schutsky et al, 2018). This is presumably attributable to the strong preference of Tet enzymes for mCG over mCA as a substrate for mC oxidation (Hu et al, 2015; DeNizio et al, 2021). Although it is possible that hmCA plays roles in gene regulation as suggested in the cerebellum (Mellén et al, 2017), it is challenging to deconvolve its roles because of our inability to exclusively eliminate this modification from the genome. We nevertheless consider that the moderate affinity of MeCP2 for this ultra-rare motif offers an unlikely basis for comprehensive new models of MeCP2 function.

# Materials and Methods

## Bioinformatic analyses

### Sequencing datasets

Table 1 below details the published datasets used for the analyses. These include chromatin immunoprecipitation followed by

**Table 1. Published datasets used for the analyses.**

| Brain tissue | Dataset | GEO accession |
|---|---|---|
| Hypothalamus | ChIP-Seq | GSE66868 (Chen et al, 2015) |
| Hypothalamus | WGBS-Seq | GSE84533 (Lagger et al, 2017) |
| Hypothalamus | RNA-Seq | GSE66870 (Chen et al, 2015) |
| Forebrain | ChIP-Seq | GSE139509 (Boxer et al, 2020) |
| Forebrain | WGBS-Seq | GSE128172 (Boxer et al, 2020) |
| Forebrain | RNA-Seq | GSE128178 (Boxer et al, 2020) |
| Frontal cortex | ChIP-Seq | GSE67293 (Gabel et al, 2015) |
| Frontal cortex | WGBS-Seq | GSE47966 (Lister et al, 2013) |
| Visual cortex | RNA-Seq | GSE67294 (Gabel et al, 2015) |

sequencing (ChIP-Seq), RNA sequencing (RNA-Seq), and bisulfite sequencing (WGBS-Seq) libraries from different regions of mouse brain quantifying MeCP2 occupancy, gene expression, and DNA methylation levels, respectively.

### RNA-Seq analyses

Gene expression analyses were performed for mouse brain RNA-Seq datasets (Table 1). Raw data were downloaded, mRNA expression was quantified using kallisto (Bray et al, 2016), and differential expression analysis was performed using DESeq2 (Love et al, 2014). Differentially regulated genes (significance threshold after Benjamini–Hochberg correction $P$-adjusted value < 0.05 or $P$-value < 0.05) in Mecp2 KO and WT mouse brain tissue were sorted according to the total amount of mCAC per gene body including [CA]$_n$ or excluding [CA]$_n$; total amount of mCA in [CA]$_n$ of gene body, binned into 30 equal-sized bins, and mean log$_2$ fold-change of each bin is plotted. Error bars represent the standard error of the mean for that bin. Alternatively, the mean log$_2$ fold-change was plotted for differentially regulated genes sorted according to the % CA repeats within the gene body and total methylation within the CA repeats.

### ChIP-Seq analyses

Raw fastq reads were downloaded from GEO (Table 1) and subsequent ChIP-Seq analysis was performed on mouse genome (mm9) using snakePipes (Bhardwaj et al, 2019) v2.5.3. log$_2$ ChIP-Seq signal over input signal and log$_2$ wild-type ChIP-Seq signal over Mecp2 KO ChIP-Seq signal where available are quantified using bigWigAverageOverBed across genomic locations containing [CA]$_n$, Repeats excluding [CA]$_n$ and rest of the genome.

### WGBS-Seq analyses

Processed WGBS-Seq datasets described in Table 1 were downloaded and DNA methylation ratios of individual cytosine nucleotides within CAC, CA, and CG contexts were determined across both the sense and non-sense strands. The whole genome is divided into 1 kb windows using bedtools, and DNA methylation is calculated in the 1 kb window labelled as mCAC/kb in Fig 2B. For repetitive regions, DNA methylation is calculated within the extended [CA]$_n$ or Repeats excluding [CA]$_n$ region to match the length of repeat regions equivalent to 1 kb. Mean DNA methylation levels for different

**Table 2.  Oligonucleotide sequences for the probes used in the pulldown assay.**

| Name | Sequence of oligonucleotide |
| --- | --- |
| C1 | 5′-B-tgcgctatgcacttgcgctatgcactttgcgctaatgcacttgcgcttattgcgcacttgcacttttgcacacgcg cacgatgcgcttaatgcgcgattgcacacgctgcacacacgcgctttgca-3′ |
| [CA]$_7$ | 5′-B-tgcgctatgcacttgcgctatgcactttgcgctaatgcacttgcgCACACACACACACActtattgcgcacttgca cttttgcacacgcgcacgatgcgcttaatgcgcgattgcacacgctgcacacacgcgctttgca-3′ |
| [CA]$_7$-1mC | 5′-B-tgcgctatgcacttgcgctatgcactttgcgctaatgcacttgcgCACAmCACACACACActtattgcgcacttgca cttttgcacacgcgcacgatgcgcttaatgcgcgattgcacacgctgcacacacgcgctttgca-3′ |
| [CA]$_7$-4mC | 5′-B-tgcgctatgcacttgcgctatgcactttgcgctaatgcacttgcgmCACAmCACAmCACAmCActtattgcgcacttg cacttttgcacacgcgcacgatgcgcttaatgcgcgattgcacacgctgcacacacgcgctttgca-3′ |
| C2 | 5′-B-tgcgctatgcacttgcgctatgcactttgcgctaatgcacttgcgCAGACAGACAGACActtattgcgca cttgcacttttgcacacgcgcacgatgcgcttaatgcgcgattgcacacgctgcacacacgcgctttgca-3′ |
| CAC | 5′-B-cgcactttgcactatgcacttgcactatgcactttgcactaatgcacttgcacttattgcacacttgca cttttgcacacacgcacgatgcacttaatgcacgattgcacacactgcacacacgcactttgcacactgca-3′ |
| mCAC | 5′-B-cgcactttgmCACtatgcacttgmCACtatgcactttgmCACtaatgcacttgmCACttattgmCACacttgcac ttttgcacamCACgcacgatgmCACttaatgmCACgattgcacamCACtgcacacacgmCACtttgcacactgca-3′ |
| CG | 5′-B-cgcactttgCGctatgcacttgCGctatgcactttgCGctaatgcacttgCGcttattgCGcacttgcacttttg cacaCGcgcacgatgCGcttaatgCGcgattgcacaCGctgcacacacgCGcttttgcacactgca-3′ |
| mCG | 5′-B-cgcactttgmCGctatgcacttgmCGctatgcactttgmCGctaatgcacttgmCGgcttattgmCGcacttgcac ttttgcacamCGcgcacgatgmCGcttaatgmCGcgattgcacamCGctgcacacacgmCGctttgcacactgca-3′ |

B, biotin, m, methyl group. All molecules were annealed to the appropriate methylated or non-methylated reverse oligonucleotide.

DNA sequence contexts are calculated for cytosines with adequate sequence coverage. For [CA]$_n$ and [CAN]$_n$, all cytosines across both strands within the genomic loci of respective repeats are considered. Because of coverage differences between the bisulfite datasets, we set the cytosine coverage thresholds for the hypothalamus (Lagger et al, 2017) and forebrain (Boxer et al, 2020) at five compared with more highly covered cortex (Lister et al, 2013) at at least 10 reads for every cytosine. These thresholds enable reliable estimates of average DNA methylation.

### [CA]$_n$, [CAN]$_n$, and Repeats excluding [CA]$_n$

The list of genomic locations containing CA and TG repeats was extracted from "Variation and Repeats" group of RepeatMasker track in mouse (mm9) genome using UCSC table browser functionality (https://genome.ucsc.edu/cgi-bin/hgTables). For humans (chm13-v1.1), RepeatMasker track was downloaded from processed data (Hoyt et al, 2022). Loci labelled "(CA)n" and "(TG)n" in the RepeatMasker track were used for [CA]$_n$. Loci labelled "(CAA)n," "(TTG)n," "(CAG)n," "(CTG)n," "(CAT)n," and "(ATG)n" were used for [CAN]$_n$. Simple repeat sequence loci other than [CA]$_n$ are considered as "Repeats excluding [CA]$_n$."

### CAC occurrences

After extracting the list of genomic loci for [CA]$_n$, CAC occurrences are calculated using bedtools and jellyfish for [CA]$_n$ and the whole mouse genome.

### Reproducibility

Source code to reproduce all the analysis and figures is available on the GitHub repository (https://github.com/kashyapchhatbar/ MeCP2_2022_manuscript) and archived at Zenodo (DOI: 10.5281/ zenodo.6997675).

### Pulldown assay for MeCP2 binding to DNA

This assay was performed as described previously (Piccolo et al, 2019) with the following modifications. Biotin end-labelled double-strand synthetic oligonucleotides (2 μg) described in Table 2 were coupled to M280-streptavidin Dynabeads according to manufacturer's instructions (Invitrogen). Bead-DNA complex was then co-incubated at 4°C for 1.5 h with nuclear protein (10 μg). Nuclear extracts from mouse brain (0.42 M salt) were prepared as described (Mellén et al, 2017) and dialysed back into a solution containing 0.15 M NaCl. After extensive washing, bead-bound proteins were eluted in Laemmli buffer (Sigma-Aldrich) and resolved on a 4–20% SDS-polyacrylamide gel (NEB). The presence of MeCP2 was assayed by Western blotting using anti-MeCP2 monoclonal antibody M6818 (Sigma-Aldrich) using IR dye as a secondary antibody (IRDye 800CW donkey anti-mouse; LI-COR Biosciences). Triplicate assays were scanned then quantified using a LI-COR Odyssey CLx machine and software.

# Supplementary Information

# Acknowledgements

We thank Raphaël Pantier and Matthew Lyst for providing critical feedback on the work. This work was supported by Wellcome Centre grant 107930/Z/ 15/Z to A Bird, who is also a member of the Simons Initiative for the Developing Brain. K Chhatbar was supported by a staff scholarship from the College of Science and Engineering, University of Edinburgh.

## Author Contributions

K Chhatbar: conceptualization, data curation, software, formal analysis, validation, investigation, visualization, methodology, and writing—original draft, review, and editing.
J Connelly: conceptualization, validation, investigation, visualization, methodology, and writing—original draft, review, and editing.
S Webb: data curation, software, formal analysis, and writing—review and editing.
S Kriaucionis: conceptualization, data curation, supervision, investigation, methodology, and writing—original draft, review, and editing.
A Bird: conceptualization, resources, supervision, funding acquisition, validation, investigation, visualization, project administration, and writing—original draft, review, and editing.

## Conflict of Interest Statement

The authors declare that they have no conflict of interest.

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
