## [Reviewer comments · Life Science Alliance]

A critique of the hypothesis that CA repeats are primary targets of neuronal MeCP2

Kashyap Chhatbar, John Connelly, Shaun Webb, Skirmantas Kriaucionis, and Adrian Bird
DOI: 10.26508/lsa.202201522

Corresponding author(s): Adrian Bird, University of Edinburgh

Review Timeline:	Submission Date:	2022-05-11
	Editorial Decision:	2022-06-17
	Revision Received:	2022-08-17
	Editorial Decision:	2022-09-06
	Revision Received:	2022-09-07
	Accepted:	2022-09-07

Scientific Editor: Novella Guidi

Transaction Report:

June 17, 2022

Re: Life Science Alliance manuscript #LSA-2022-01522-T

Prof. Adrian P. Bird
Edinburgh, University of
The Wellcome Trust Centre for Cell Biology
The King's Buildings
West Mains Road
UK-Edinburgh EH9 3BF, University of Edinburgh The King's Buildings
United Kingdom

Dear Dr. Bird,

Thank you for submitting your manuscript entitled "Evaluation of CA repeat arrays as targets for MeCP2 function in the brain" to Life Science Alliance. The manuscript was assessed by expert reviewers, whose comments are appended to this letter. We invite you to submit a revised manuscript addressing the Reviewer comments.

Thank you for this interesting contribution to Life Science Alliance. We are looking forward to receiving your revised manuscript.

Sincerely,

B. MANUSCRIPT ORGANIZATION AND FORMATTING:

Reviewer #1 (Comments to the Authors (Required)):

In the current study the authors provide compelling evidence that MeCP2 does not locate preferentially to CA repeats. MeCP2 is the causal gene for RETT syndrome and an established binder of methylated cytosines in the CG and CA context.

The current analysis carefully reviews existing datasets and perform in vitro experiments to show that this binding does not extend to unmodified CA.

This is a relevant finding particularly in light of recent claims, published in Science, that MeCP2 binds CA repeats and I fully support publication.

Reviewer #2 (Comments to the Authors (Required)):

This study from Chhatbar et al investigates the conclusions made in a recent publication (Ibrahim et al, Science 2022) and calls into question whether CA repeat sequences are the primary targets of the DNA methylation reader protein methyl-CpG-binding protein 2 (MeCP2), mutations of which cause Rett syndrome. The authors re-analyzed a number of relevant datasets generated using mouse models of Rett syndrome across multiple brain regions and performed pulldown assays with appropriate controls to investigate the binding between MeCP2 and DNA. They find that cytosine methylation is not enriched in CA repeats in the brain across multiple datasets, nor do they find evidence of preferential MeCP2 binding at CA repeats, but instead detect enrichment of MeCP2 binding at methylated cytosines in the CAC context. The latter is consistent with previous findings from their lab and others. Additionally, the authors demonstrate that MeCP2 binds poorly to unmethylated CA repeats using a pulldown assay. Finally, the authors analyze multiple RNA-seq datasets to demonstrate only a limited correlation between CA repeat density and MeCP2-dependent gene regulation.

Overall, this manuscript provides strong evidence to counter the conclusions drawn from the Ibrahim et al study. Given that MeCP2 is predominantly expressed in neurons of the central nervous system and that Rett syndrome is a neurological disorder, a careful characterization of the relationship between MeCP2 and CA repeats in relevant tissues and cell types is warranted. This is in contrast to the Ibrahim et al study, which primarily used mouse embryonic fibroblast cells. Findings presented in this manuscript are well-controlled and convincing, and would be acceptable for publishing. Several comments, as outlined below, if appropriately addressed, would improve the study.

Major Points:

1. Given the importance of subtleties in the analysis of next-generation sequencing data, the code used to reproduce findings and figures from this manuscript should be made publicly available, ideally in an appropriate environment such as GitHub.

2. The authors should consider placing more emphasis on the paucity of 5hmCA in the brain, as this is a large caveat to the conclusions from the Ibrahim et al study. Ibrahim et al used the Mellen et al dataset from the cerebellum to make claims about hmCA in the brain. Other brain regions or cell types, such as cortical projection neurons, have scarce amounts of this particular modification (Schutsky et al Nature Biotech 2018, Lister et al Science 2013). Furthermore, deleting MeCP2 from the cerebellum resulted in minimal phenotypes (Achilly et al eLife 2021).

3. The authors may tone down conclusions about the lack of underrepresentation of CAC trinucleotides inside of CA repeats in WGBS datasets. In Figure 1D, there is wide variability across the different datasets, but almost all of them display underrepresentation. An approximate 2-fold decrease in representation is still a large decrease, and the conclusions should be changed to better reflect the data presented.

4. A few details related to data processing are lacking and need to be described further in the methods. First, it is unclear whether the asymmetric nature of non-CG methylation was appropriately considered. The coordinates for CA repeats (using (CA)_n and (TG)_n simple repeats from Repeatmasker) seem to have been attained appropriately, but it is unclear if the correct strand was used for mCA analysis (only the + strand at CA sequences and only the - strand over TG sequences). Given the

asymmetric nature of non-CG methylation, strand-specificity is a critical aspect of the analysis that should be described in further detail. Additionally, more details about the "nonCA repeats" should be included: the text states that "other repeat sequence loci are considered as nonCA repeats" - were these all other simple repeats that weren't the ones described, or did this include other types of repetitive elements?

5. In the Ibrahim et al study, much of the argument for MeCP2 enrichment at CA repeats came from enrichment plots of MeCP2 ChIP-seq centered over CA repeats using frontal cortex data from Gabel et al 2015 and Lister et al 2013. This study would benefit from a response to this. Enrichment plots can be misleading and dependent on the length of the surrounding region plotted, and specific magnitudes of enrichment can be hard to compare. This may be particularly important in the context of the genome-wide distribution of non-CG methylation. Especially considering that the current study found some surprising results with the matched Gabel et al and Lister et al datasets, it would be informative to repeat these enrichment analyses across multiple datasets, as was done in other parts of the current study.

6. The focus on mCAC opposed to mCA is warranted in some analyses, but not others. Previous work from the authors has convincingly demonstrated that MeCP2 affinity and occupancy is significantly higher for the mCAC motif compared to the mCAD motif, but the data are not yet definitive enough to exclude mCAD motifs as non-relevant. The authors may break down the data by analyzing all mCA, mCAC, and mCAD separately. However, for analyzing enrichment of DNA methylation within CA repeats vs. outside of the repeats, only the CAC trinucleotide should be considered (as the authors have correctly done in the current version of the manuscript) to ensure an apples-to-apples comparison is made between CA within repeats (which is by definition in the CAC context) and CA outside of repeats (which is a mix of CAC and CAD contexts), given that the CAC trinucleotide is methylated by DNMT3A at much higher levels in neurons than other CAN trinucleotides. The authors may explain this in more detail in the text.

7. There may be other ways to present the data in Figure 4 that would strengthen the argument that the strongest signal is between mCAC and gene expression and not the presence of CA repeats. For example, boxplots comparing CA repeat density of up- and down-DEGs to all expressed genes, DEGs from other datasets, etc. may be useful to strengthen this part of the manuscript.

8. Given the limitations of profiling gene expression from bulk tissues (or sorted neurons using a pan-neuronal marker), which is confounded by the presence of post-transcriptional regulation and cell type heterogeneity, it would notably strengthen the study to perform the same type of analysis with cell type-specific nuclear RNA-seq data, such as the data from Johnson et al 2017 with DNA methylation data from the same excitatory neurons (WGBS data from Mo et al 2015).

Minor Points:

1. A justification for different read coverage thresholds for different WGBS datasets would be helpful to include. Assuming it has to do with the different depths the samples were sequenced at.

2. To make the analysis presented in Fig. 1C convincing, it would be helpful to also analyze this trend genome-wide (not just at genes) by binning the genome into small bins and computing the number of methylated CA's within and outside of CA repeats across the bins.

3. What does the "-" stand for in Fig. 3B? Unmethylated?

4. Figure 4: was frontal cortex or visual cortex RNA-seq data used in this figure?

Reviewer #3 (Comments to the Authors (Required)):

The manuscript by Chhatbar et al. investigates a recent claim that methylated and hydroxymethylated CA repeat arrays are the primary sites for MeCP2 binding and function (Ibrahim et al., Science 2021). The findings from Ibrahim et al differed from prior findings from multiple labs that methylated CAC and CG are the primary sites for MeCP2 binding and function. Ibrahim et al. performed most of their experiments in fibroblasts, but MeCP2 is most highly expressed in neurons, so Chhatbar et al carefully analyzed several published brain datasets to test the relevance and validity of the findings from Ibrahim et al. Several key findings from Chhatbar et al do not support the conclusions from Ibrahim et al, including: that cytosine methylation is not more enriched in CA repeats than in CAC sequences outside repeats, that MeCP2 does not bind preferentially in CA repeats in the brain or in vitro, and that the gene misregulation associated with loss of MeCP2 correlates with mCAC levels rather than the percentage of CA repeats in genes. Overall, these findings suggest that CA repeats are not the primary site for MeCP2 binding and function in the brain, rather, mCAC and mCG are the primary sites for the MeCP2 binding and function. This study represents an important advance for the field because the claim that CA repeats are the primary target for MeCP2 function had the potential to greatly influence the current understanding of MeCP2 function and Rett syndrome, especially since the paper was published in a high-impact journal. It is very important for the field that these findings be independently and carefully tested in the relevant cell type. I have a few suggestions below to strengthen the analyses that support the key conclusions in this

study.

In Figure 1, the authors find that cytosine methylation is not more enriched in CA repeat regions than in CAC sequences outside repeats in the brain. This finding is well supported by re-analysis of three published brain bisulfite sequencing datasets. They also point out that Ibrahim et al did not consider that mCA occurs almost exclusively in the mCAC context, and thus mCA levels outside of repeats must be analyzed in the mCAC context. They also address the claim from Ibrahim et al that CA repeats are underrepresented in bisulfite sequencing data in Figure 1D. However, in 3/5 datasets studied, the %CAC covered is about two-fold lower in CA repeats than the rest of the genome, and in 1 dataset (FC whole) %CAC coverage is drastically reduced in CA repeats. The authors should consider whether the reduced coverage in CA repeats in certain datasets may influence any of the analysis throughout the manuscript, and if it doesn't, should include a sentence or two explaining that.

In Figure 2, the authors find that MeCP2 binding is not enriched at CA repeat arrays in the brain. These findings are supported by re-analysis of three independent published MeCP2 ChIP-seq datasets from different brain regions. I have a few suggestions for analyses and modifications to further support this finding:

1. In Fig 2 C, D, and E, the authors plot MeCP2 ChIP signal in bins of increasing mCAC levels for regions in CA repeats, non-CA repeats, or the genome minus CA repeats. While the patterns appear similar in all 3 genomic regions, there is no statistical analysis. It would strengthen these results to include the correlation values between MeCP2 ChIP and mCAC in each type of genomic regions.
2. In Fig 2E, the authors find that there is no correlation between MeCP2 binding and mCAC for the frontal cortex data (Gabel et al. 2015 and Lister et al. 2013). This is a confusing finding, given that Gabel et al. 2015 reported a genome-wide correlation between MeCP2 binding and mCA levels. The authors should comment further on this discrepancy—is the difference analyzing mCAC vs. mCA, or is it something about how the analysis was done?
3. Fig 2 C, D, and E suggest that MeCP2 binding correlates with mCAC regardless of whether CAC is in a repeat or not. However, there is no analysis of whether MeCP2 binding is enriched in CA repeats regardless of methylation level. In Fig 2C, MeCP2 ChIP/input in CA[n] appears to be slightly higher overall than genome minus CA[n]. The authors should more formally quantify whether MeCP2 binding is higher in CA[n] versus genome minus CA[n] in bins of the same level of mCAC.

In Figure 3, the authors perform biotinylated DNA pulldown experiments from brain extracts and find that MeCP2 binding is strengthened by adding more mCAC, either in the context of CA repeats or isolated mCAC. The results generally support this conclusion, and the authors acknowledge that the relationship between MeCP2 binding and mCAC versus CA repeats is not quantitatively proven by their data. This is presumably because the [CA]7 probe with 4 mCs appears to bind MeCP2 as strongly as the CAC probe with 10 mCs, and there is no direct comparison of [CA]7 to CAC with the same number of mCs. The authors could consider adding additional pull downs that compare [CA]7 to CAC probes with the same number of mCs to further strengthen this conclusion, or at least more specifically state why this relationship is not quantitatively proven with their data and what the limitations are.

In Figure 4, the authors find that the gene misregulation in MeCP2 KO vs. WT does not correlate with the percentage of CA repeats within gene bodies, but instead correlates with the number of mCAC per gene, regardless of whether CA repeats are included or excluded. I have a few suggestions to strengthen this analysis to support the conclusions:

1. Gene misregulation in the absence of MeCP2 has been shown to correlate with the length of genes. The percentage of CA repeats within the gene body is normalized for the length of genes, but the number of mCACs per gene does not appear to be normalized for the length of genes. The authors should normalize the number of mCACs to gene length for this analysis to verify that the stronger correlation observed for mCAC than %[CA]n is not purely due to the correlation of gene misregulation with gene length.
2. It would be helpful to clarify whether the genes shown in Fig 4 are significantly misregulated or all expressed genes, and to indicate how many genes were significantly up and down regulated in each dataset analyzed.
3. One possible explanation for the lack of correlation between gene misregulation and %[CA]n is that, like in Ibrahim et al, both the up and down regulated genes have high levels of CA repeats. It would be good to include boxplots of the %[CA]n in up, down, and not regulated genes like in Ibrahim et al, to investigate whether this is the case.

#LSA-2022-01522-T revision**Chhatbar et al****A critique of the hypothesis that CA repeats are primary targets of neuronal MeCP2****Reviewer #1 (Comments to the Authors (Required)):**

In the current study the authors provide compelling evidence that MeCP2 does not locate preferentially to CA repeats. MeCP2 is the causal gene for RETT syndrome and an established binder of methylated cytosines in the CG and CA context.

The current analysis carefully reviews existing datasets and perform in vitro experiments to show that this binding does not extend to unmodified CA. This is a relevant finding particularly in light of recent claims, published in Science, that MeCP2 binds CA repeats and I fully support publication.

The reviewer's positive comments and support for publication are appreciated.

Reviewer #2 (Comments to the Authors (Required)):

This study from Chhatbar et al investigates the conclusions made in a recent publication (Ibrahim et al, Science 2022) and calls into question whether CA repeat sequences are the primary targets of the DNA methylation reader protein methyl-CpG-binding protein 2 (MeCP2), mutations of which cause Rett syndrome. The authors re-analyzed a number of relevant datasets generated using mouse models of Rett syndrome across multiple brain regions and performed pulldown assays with appropriate controls to investigate the binding between MeCP2 and DNA. They find that cytosine methylation is not enriched in CA repeats in the brain across multiple datasets, nor do they find evidence of preferential MeCP2 binding at CA repeats, but instead detect enrichment of MeCP2 binding at methylated cytosines in the CAC context. The latter is consistent with previous findings from their lab and others. Additionally, the authors demonstrate that MeCP2 binds poorly to unmethylated CA repeats using a pulldown assay. Finally, the authors analyze multiple RNA-seq datasets to demonstrate only a limited correlation between CA repeat density and MeCP2-dependent gene regulation.

Overall, this manuscript provides strong evidence to counter the conclusions drawn from the Ibrahim et al study. Given that MeCP2 is predominantly expressed in neurons of the central nervous system and that Rett syndrome is a neurological disorder, a careful characterization of the relationship between MeCP2 and CA repeats in relevant tissues and cell types is warranted. This is in contrast to the Ibrahim et al study, which primarily used mouse embryonic fibroblast cells. Findings presented in this manuscript are well-controlled and convincing, and would be acceptable for publishing. Several comments, as outlined below, if appropriately addressed, would improve the study.

The reviewer's critical reading and suggestions are appreciated. We have responded to individual comments below.

Major Points:

1. Given the importance of subtleties in the analysis of next-generation sequencing data, the code used to reproduce findings and figures from this manuscript should be made publicly available, ideally in an appropriate environment such as GitHub.

Code needed to reproduce the findings and figures is included in the Methods section (page 12, lines 325-327) of the revised manuscript. The link to the github repository is https://github.com/kashyapchhatbar/MeCP2_2022_manuscript

2. The authors should consider placing more emphasis on the paucity of 5hmCA in the brain, as this is a large caveat to the conclusions from the Ibrahim et al study. Ibrahim et al used the Mellen et al dataset from the cerebellum to make claims about hmCA in the brain. Other brain regions or cell types, such as cortical projection neurons, have scarce amounts of this particular modification (Schutsky et al Nature Biotech 2018,

Lister et al Science 2013). Furthermore, deleting MeCP2 from the cerebellum resulted in minimal phenotypes (Achilly et al eLife 2021).

We have added to the discussion of hmCA rarity (pages 9-10; lines 262-267). In our view, while the cerebellar-specific Mecp2 KO phenotype demonstrates that Mecp2 in that brain region plays a relatively minor role in the burden of Rett-like phenotype, the available data does not permit the conclusion that this is attributable to hmCA abundance.

3. The authors may tone down conclusions about the lack of underrepresentation of CAC trinucleotides inside of CA repeats in WGBS datasets. In Figure 1D, there is wide variability across the different datasets, but almost all of them display underrepresentation. An approximate 2-fold decrease in representation is still a large decrease, and the conclusions should be changed to better reflect the data presented.

We re-visited the processed WGBS data in order to check the coverage of individual cytosines. Methylation levels were as in the original version, but coverage showed minor variation as shown in revised Figure 1D. The source code used is provided in the latest version of the manuscript (page 12, lines 325-327). Representation of CAC in CA repeats compared to the rest of the genome is in fact significantly less than 2-fold in all cases with the striking exception of FC whole (Figure 1D). We have expanded the explanation of CA repeat coverage in the bisulfite datasets to make clearer our interpretation (pages 3-4, lines 79-99). We explain why the small reduction in WGBS coverage of some datasets does not significantly affect our conclusion that mCA is approximately as frequent within and outside repeat arrays. This is because the number of repeat arrays with sufficient coverage is large enough to provide statistically robust estimates for mCA frequency within CA repeats. Finally, we point out that the whole frontal cortex dataset (FC whole) is an outlier due to its much lower repeat coverage than any of the other WGBS datasets. Importantly, this was the only dataset used by Ibrahim et al for their analysis and based on this atypical data, they inferred – incorrectly we suggest - that CA repeats are intrinsically under-represented by WGBS.

4. A few details related to data processing are lacking and need to be described further in the methods. First, it is unclear whether the asymmetric nature of non-CG methylation was appropriately considered. The coordinates for CA repeats (using (CA)_n and (TG)_n simple repeats from Repeatmasker) seem to have been attained appropriately, but it is unclear if the correct strand was used for mCA analysis (only the + strand at CA sequences and only the - strand over TG sequences). Given the asymmetric nature of non-CG methylation, strand-specificity is a critical aspect of the analysis that should be described in further detail. Additionally, more details about the "nonCA repeats" should be included: the text states that "other repeat sequence loci are considered as nonCA repeats" - were these all other simple repeats that weren't the ones described, or did this include other types of repetitive elements?

Our analysis includes all cytosines on both strands and subsequently filtered according to the di/tri-nucleotide context and therefore takes into account the asymmetric nature of non-CG methylation. Regarding nonCA repeats now labelled "Repeats excluding [CA]_n", we only considered "simple repeats" other than [CA]_n which are defined by RepBase and collated by Repeat-Masker. This information is presented in our updated Methods section (page 12, lines 311-320).

5. In the Ibrahim et al study, much of the argument for MeCP2 enrichment at CA repeats came from enrichment plots of MeCP2 ChIP-seq centered over CA repeats using frontal cortex data from Gabel et al 2015 and Lister et al 2013. This study would benefit from a response to this. Enrichment plots can be misleading and dependent on the length of the surrounding region plotted, and specific magnitudes of enrichment can be hard to compare. This may be particularly important in the context of the genome-wide distribution of non-CG methylation. Especially considering that the current study found some surprising results with the matched

Gabel et al and Lister et al datasets, it would be informative to repeat these enrichment analyses across multiple datasets, as was done in other parts of the current study.

We agree with the reviewer's reservations about enrichment analysis. Our efforts to replicate the analyses of Ibrahim et al using this approach were unsuccessful. Based on their Fig 4E and the Methods supplied, we deduce that the profile plots for "inside CAn" are derived from an average from ~300,000 loci while the "outside CAn" plots are based on several million loci. This large disparity in the number of loci between different categories, together with the fact that these plots do not account for levels of DNA methylation and use "single mean value" comparisons for multiple categories (CAn, outside CAn, mCG etc), makes robust interpretation difficult.

Recognizing the limitations of enrichment plots, we chose an alternative approach that takes into account levels of DNA methylation as shown in Figure 2C-E of the revised manuscript. The new analysis clearly shows the relationship between MeCP2 binding and mCAC (page 5, lines 123-138). We are not convinced that extending enrichment analysis to multiple datasets and perhaps entering into a critique of the enrichment approach would be a useful exercise in this manuscript. While we accept that combining the Gabel and Lister frontal cortex datasets showed only modest MeCP2 enrichment over highly methylated regions compared with the other datasets (see revised Figure 2E), we have no reason to believe that this is linked to the use of enrichment plots. Instead, we consider that while the WGBS data for cortex achieves high coverage, the resolution of this particular ChIP dataset is comparatively low, as suggested by the greatly reduced number of MeCP2 peaks (page 6, lines 142-148). The contrast between this ChIP profile compared with the closely similar profiles obtained by Boxer et al for forebrain (which of course includes frontal cortex) and by Chen et al for hypothalamus is striking (see Figure 2B). The fact that two different laboratories studying different brain regions obtained closely matching MeCP2 ChIP patterns appears to confirm that the frontal cortex dataset is an outlier.

6. The focus on mCAC opposed to mCA is warranted in some analyses, but not others. Previous work from the authors has convincingly demonstrated that MeCP2 affinity and occupancy is significantly higher for the mCAC motif compared to the mCAD motif, but the data are not yet definitive enough to exclude mCAD motifs as non-relevant. The authors may break down the data by analyzing all mCA, mCAC, and mCAD separately. However, for analyzing enrichment of DNA methylation within CA repeats vs. outside of the repeats, only the CAC trinucleotide should be considered (as the authors have correctly done in the current version of the manuscript) to ensure an apples-to-apples comparison is made between CA within repeats (which is by definition in the CAC context) and CA outside of repeats (which is a mix of CAC and CAD contexts), given that the CAC trinucleotide is methylated by DNMT3A at much higher levels in neurons than other CAn trinucleotides. The authors may explain this in more detail in the text.

We agree that our analysis focusses on mCAC and ignores mCAD, whose biological significance is unclear. The reason, as acknowledged by the reviewer, is that only CAC is found within CA repeats (while mCAD is not) and CAC is by far the most frequent target of non-CG methylation in the brain. While we accept that some biological importance of mCAD has not been ruled out, its absence from CA repeats, which are the main focus of this study, would distract from the theme of our paper. We are not aware of an obvious hypothesis that would be tested by this analysis. These considerations mean that we prefer not to include these rare modified sites in our study.

7. There may be other ways to present the data in Figure 4 that would strengthen the argument that the strongest signal is between mCAC and gene expression and not the presence of CA repeats. For example, boxplots comparing CA repeat density of up- and down-DEGs to all expressed genes, DEGs from other datasets, etc. may be useful to strengthen this part of the manuscript.

We have added boxplots comparing CA repeat density of up- and down-regulated genes with all expressed genes (new supplementary Figure S1A, page 7, lines 178-182) and saw no correlation. The apparent absence of [CA]n

enrichment in up-regulated or down-regulated versus unchanged genes emphasises that gene mis-regulation in the absence of MeCP2 is independent of CA repeat density, further strengthening the conclusions of our manuscript.

8. Given the limitations of profiling gene expression from bulk tissues (or sorted neurons using a pan-neuronal marker), which is confounded by the presence of post-transcriptional regulation and cell type heterogeneity, it would notably strengthen the study to perform the same type of analysis with cell type-specific nuclear RNA-seq data, such as the data from Johnson et al 2017 with DNA methylation data from the same excitatory neurons (WGBS data from Mo et al 2015).

Cell-type specific RNA seq on specific neuronal subtypes might be a useful method of addressing this issue further, but in our opinion this significant additional experimental work would be more justified if the analysis of brain regions in Figure 4 encouraged the belief that CA-repeats were key drivers of MeCP2 function. On the contrary, Figure 4 shows that whereas we see convincing correlations between mCAC content of genes and mRNA up-regulation when MeCP2 is absent, no comparable correlation with CA repeats was detected. A strength of the present study is provided by the coherence of findings across parallel datasets from different research groups. To our knowledge only one RNA-seq + WGBS dataset from excitatory neurons is available (see Referee's suggestion), so the comparative aspect is not available without considerable further work. Since all the data in our manuscript places the importance of CA repeats in serious question, we do not consider that the extra time and expense is justified at this stage.

Minor Points:

1. A justification for different read coverage thresholds for different WGBS datasets would be helpful to include. Assuming it has to do with the different depths the samples were sequenced at.

The reviewer's assumption is correct, and we have rectified this omission by including this information in our Methods section (page 11, lines 307-310).

2. To make the analysis presented in Fig. 1C convincing, it would be helpful to also analyze this trend genome-wide (not just at genes) by binning the genome into small bins and computing the number of methylated CA's within and outside of CA repeats across the bins.

We have added an extra row in Figure 1B which shows average methylation levels in CAC across the genome excluding CA repeats. This re-enforces our conclusion that [CA]_n arrays acquire DNA methylation levels comparable to the surrounding DNA.

3. What does the "-" stand for in Fig. 3B? Unmethylated?

Yes – this is now stated in the legend to Figure 3.

4. Figure 4: was frontal cortex or visual cortex RNA-seq data used in this figure?

According to the description in Gabel et al., RNA-seq is performed on visual cortex tissue from mice. We have re-named "visual cortex" in the figure panels as "Cortex" to avoid confusion.

Reviewer #3 (Comments to the Authors (Required)):

The manuscript by Chhatbar et al. investigates a recent claim that methylated and hydroxymethylated CA repeat arrays are the primary sites for MeCP2 binding and function (Ibrahim et al., Science 2021). The findings from Ibrahim et al differed from prior findings from multiple labs that methylated CAC and CG are the primary sites for MeCP2 binding and function. Ibrahim et al. performed most of their experiments in fibroblasts, but MeCP2 is most highly expressed in neurons, so Chhatbar et al carefully analyzed several published brain datasets to test the relevance and validity of the findings from Ibrahim et al. Several key findings from Chhatbar et al do not support the conclusions from Ibrahim et al, including: that cytosine methylation is not more enriched in CA repeats than in CAC sequences outside repeats, that MeCP2 does not bind preferentially in CA repeats in the brain or in vitro, and that the gene misregulation associated with loss of MeCP2 correlates with mCAC levels rather than the percentage of CA repeats in genes. Overall, these findings suggest that CA repeats are not the primary site for MeCP2 binding and function in the brain, rather, mCAC and mCG are the primary sites for the MeCP2 binding and function. This study represents an important advance for the field because the claim that CA repeats are the primary target for MeCP2 function had the potential to greatly influence the current understanding of MeCP2 function and Rett syndrome, especially since the paper was published in a high-impact journal. It is very important for the field that these findings be independently and carefully tested in the relevant cell type. I have a few suggestions below to strengthen the analyses that support the key conclusions in this study.

We welcome that the reviewer finds our manuscript pertinent. We have responded to individual comments below.

In Figure 1, the authors find that cytosine methylation is not more enriched in CA repeat regions than in CAC sequences outside repeats in the brain. This finding is well supported by re-analysis of three published brain bisulfite sequencing datasets. They also point out that Ibrahim et al did not consider that mCA occurs almost exclusively in the mCAC context, and thus mCA levels outside of repeats must be analyzed in the mCAC context. They also address the claim from Ibrahim et al that CA repeats are underrepresented in bisulfite sequencing data in Figure 1D. However, in 3/5 datasets studied, the %CAC covered is about two-fold lower in CA repeats than the rest of the genome, and in 1 dataset (FC whole) %CAC coverage is drastically reduced in CA repeats. The authors should consider whether the reduced coverage in CA repeats in certain datasets may influence any of the analysis throughout the manuscript, and if it doesn't, should include a sentence or two explaining that.

This point was also raised by reviewer 2 and our response is repeated here. We re-visited the processed WGBS data in order to check the coverage of individual cytosines. Methylation levels were as in the original version, but coverage showed minor variation as shown in revised Figure 1D (source code used is provided in the latest version of the manuscript, page 12 lines 324-327). Representation of CAC in CA repeats compared to the rest of the genome is in fact significantly less than 2-fold in all cases with the striking exception of FC whole (Figure 1D). We have expanded the explanation of CA repeat coverage in the bisulfite datasets to make clearer our interpretation (pages 3-4, lines 79-99). We explain why the small reduction in WGBS coverage of some datasets does not significantly affect our conclusion that mCA is approximately as frequent within and outside repeat arrays. This is because the number of repeat arrays with sufficient coverage is large enough to provide statistically robust estimates for mCA frequency within CA repeats. Finally, we point out that the whole frontal cortex dataset (FC whole) is an outlier due to its much lower repeat coverage than any of the other WGBS datasets. Importantly, this was the only dataset used by Ibrahim et al for their analysis and based on this atypical data, they inferred – incorrectly we suggest - that CA repeats are intrinsically under-represented by WGBS.

In Figure 2, the authors find that MeCP2 binding is not enriched at CA repeat arrays in the brain. These findings are supported by re-analysis of three independent published MeCP2 ChIP-seq datasets from different brain regions. I have a few suggestions for analyses and modifications to further support this finding:

1. In Fig 2 C, D, and E, the authors plot MeCP2 ChIP signal in bins of increasing mCAC levels for regions in CA repeats, non-CA repeats, or the genome minus CA repeats. While the patterns appear similar in all 3 genomic regions, there is no statistical analysis. It would strengthen these results to include the correlation values between MeCP2 ChIP and mCAC in each type of genomic regions.

We have revised our analysis to include the rank correlation between bins of MeCP2 enrichment and mCAC methylation to make the data in Figure 2C, 2D and 2E statistically robust.

2. In Fig 2E, the authors find that there is no correlation between MeCP2 binding and mCAC for the frontal cortex data (Gabel et al. 2015 and Lister et al. 2013). This is a confusing finding, given that Gabel et al. 2015 reported a genome-wide correlation between MeCP2 binding and mCA levels. The authors should comment further on this discrepancy—is the difference analyzing mCAC vs. mCA, or is it something about how the analysis was done?

We agree that this discrepancy needed further explanation and have therefore updated the analysis and presented the data in graphical form in new Figure 2C-E. Using binned mean values rather than box plots and median values, we now see a modest correlation between MeCP2 enrichment and mCAC levels (new Figure 2E) as reported previously. We have altered the text (pages 5-6, lines 135-151) accordingly.

3. Fig 2 C, D, and E suggest that MeCP2 binding correlates with mCAC regardless of whether CAC is in a repeat or not. However, there is no analysis of whether MeCP2 binding is enriched in CA repeats regardless of methylation level. In Fig 2C, MeCP2 ChIP/input in CA[n] appears to be slightly higher overall than genome minus CA[n]. The authors should more formally quantify whether MeCP2 binding is higher in CA[n] versus genome minus CA[n] in bins of the same level of mCAC.

We have replotted this data in a way that reveals the granularity of the analysis and shows that the number of datapoints at high mCAC is small and consequently somewhat more variable. The plots of [CA]n versus genome-minus-[CA]n are almost superimposable. In our opinion, this data offers no support for the notion that [CA]n is a preferred target for MeCP2 binding. We conclude that mCAC is the primary influence on MeCP2 binding, regardless of its location.

In Figure 3, the authors perform biotinylated DNA pulldown experiments from brain extracts and find that MeCP2 binding is strengthened by adding more mCAC, either in the context of CA repeats or isolated mCAC. The results generally support this conclusion, and the authors acknowledge that the relationship between MeCP2 binding and mCAC versus CA repeats is not quantitatively proven by their data. This is presumably because the [CA]7 probe with 4 mCs appears to bind MeCP2 as strongly as the CAC probe with 10 mCs, and there is no direct comparison of [CA]7 to CAC with the same number of mCs. The authors could consider adding additional pull downs that compare [CA]7 to CAC probes with the same number of mCs to further strengthen this conclusion, or at least more specifically state why this relationship is not quantitatively proven with their data and what the limitations are.

These experiments failed to detect an intrinsic affinity between MeCP2 and non-methylated [CA]7 (as proposed by Ibrahim et al). Addition of one mC greatly increased binding, but increasing this further to 4 mCACs appeared to enhance binding still further (which would not fit with the cooperative model proposed by Ibrahim et al). These were the most important conclusions from this data. However, we accept that our original version failed to discuss the uncertainties inherent in some aspects of these results. We did not acknowledge the absence of a further enhancement of binding in the probe with 10 mCACs compared to 4. We recognise that there are several possible explanations for this result (e.g. steric hindrance between multiple proteins bound to single short oligonucleotide) but did not explore this issue further due to intrinsic variability of this assay. We have now included discussion of these limitations in the Results section (page 6, lines 169-173).

In Figure 4, the authors find that the gene mis-regulation in MeCP2 KO vs. WT does not correlate with the percentage of CA repeats within gene bodies, but instead correlates with the number of mCAC per gene, regardless of whether CA repeats are included or excluded. I have a few suggestions to strengthen this analysis to support the conclusions:

1. Gene mis-regulation in the absence of MeCP2 has been shown to correlate with the length of genes. The percentage of CA repeats within the gene body is normalized for the length of genes, but the number of mCACs per gene does not appear to be normalized for the length of genes. The authors should normalize the number of mCACs to gene length for this analysis to verify that the stronger correlation observed for mCAC than %[CA]_n is not purely due to the correlation of gene misregulation with gene length.

We followed this helpful suggestion to see if the correlation between gene misregulation and mCAC persists when this is expressed as % of gene length (Figure S1B). The correlation was still evident, confirming that the relationship is not attributable to gene length.

2. It would be helpful to clarify whether the genes shown in Fig 4 are significantly mis-regulated or all expressed genes, and to indicate how many genes were significantly up and down regulated in each dataset analyzed.

As requested, we have added the number of significantly regulated genes to the figures (Figure 4 and Figure S1A).

3. One possible explanation for the lack of correlation between gene misregulation and %[CA]_n is that, like in Ibrahim et al, both the up and down regulated genes have high levels of CA repeats. It would be good to include boxplots of the %[CA]_n in up, down, and not regulated genes like in Ibrahim et al, to investigate whether this is the case.

See also point 7 of Reviewer 2. We have added boxplots comparing CA repeat density of up- and down-regulated genes with unchanged genes (new supplementary Figure S1A; page 7 lines 178-182) and in each dataset found very similar levels. The apparent absence of [CA]_n enrichment in up-regulated or down-regulated versus unchanged genes suggests that gene mis-regulation in the absence of MeCP2 is independent of CA repeat density, further strengthening the conclusions of the manuscript.

September 6, 2022

RE: Life Science Alliance Manuscript #LSA-2022-01522-TR

Dr. Adrian Bird
University of Edinburgh
University of Edinburgh
Dept. of Cell & Molecular Biology
Edinburgh, Sc EH9 3JR
United Kingdom

Dear Dr. Bird,

Thank you for submitting your revised manuscript entitled "A critique of the hypothesis that CA repeats are primary targets of neuronal MeCP2". We would be happy to publish your paper in Life Science Alliance pending final revisions necessary to meet our formatting guidelines.

- please address the remaining Reviewer 2 and 3' points
- please use the [10 author names, et al.] format in your references (i.e. limit the author names to the first 10)

A. FINAL FILES:

B. MANUSCRIPT ORGANIZATION AND FORMATTING:

Sincerely,

Reviewer #2 (Comments to the Authors (Required)):

In this revision, the authors have addressed most of the concerns raised in the first round of review. Regarding MeCP2 ChIP-seq signal over the Lister frontal cortex WGBS vs. Gabel frontal cortex data, the new analysis still does not show much enrichment of MeCP2 ChIP signal over highly-methylated regions in the mCAC context - perhaps this analysis should be repeated with the same technique with the Gabel et al cortex WGBS and frontal cortex ChIP-seq datasets to see if this finding is specific to the analysis method (vs. that of Gabel et al).

Reviewer #3 (Comments to the Authors (Required)):

The authors have thoroughly addressed all my comments, and the manuscript is now suitable for publication.

My only remaining minor suggestion is to add statistical analysis to the box plots in the new Figure S1A.

Reviewer #2 (Comments to the Authors (Required)):

In this revision, the authors have addressed most of the concerns raised in the first round of review. Regarding MeCP2 ChIP-seq signal over the Lister frontal cortex WGBS vs. Gabel frontal cortex data, the new analysis still does not show much enrichment of MeCP2 ChIP signal over highly-methylated regions in the mCAC context - perhaps this analysis should be repeated with the same technique with the Gabel et al cortex WGBS and frontal cortex ChIP-seq datasets to see if this finding is specific to the analysis method (vs. that of Gabel et al).

Gabel et al profiled the MeCP2 ChIP-seq signal within gene bodies whereas our analysis considers ChIP-seq signal and total mCAC levels in 1kb windows across the whole genome. Because the resolution of ChIP-seq is a few hundred base pairs, we consider that quantifying the signal across 1kb windows provides a robust genome-wide measure within the limits of this dataset (see Figure 2C & 2D). We cannot see how repeating the analysis of Gabel et al on the low-coverage cortex WGBS (GSE60062) dataset would impact the conclusions of our paper. Therefore, in our opinion this is not a useful exercise.

Reviewer #3 (Comments to the Authors (Required)):

The authors have thoroughly addressed all my comments, and the manuscript is now suitable for publication. My only remaining minor suggestion is to add statistical analysis to the box plots in the new Figure S1A.

We omitted measures of significance in Figure S1A as the large numbers involved automatically generate differences that register as "significant". The point of the figure is to illustrate the tiny magnitude of the changes (e.g. 0.6% versus 0.78%). Whether or not they are "significant" is much less relevant and something of a distraction in our opinion. Brackets/stars could of course be added but our preference is to leave the figure as it is.

September 7, 2022

RE: Life Science Alliance Manuscript #LSA-2022-01522-TRR

Dr. Adrian Bird
University of Edinburgh
University of Edinburgh
Dept. of Cell & Molecular Biology
Edinburgh, Sc EH9 3JR
United Kingdom

Dear Dr. Bird,

Thank you for submitting your Research Article entitled "A critique of the hypothesis that CA repeats are primary targets of neuronal MeCP2". It is a pleasure to let you know that your manuscript is now accepted for publication in Life Science Alliance. Congratulations on this interesting work.

DISTRIBUTION OF MATERIALS:

Again, congratulations on a very nice paper. I hope you found the review process to be constructive and are pleased with how the manuscript was handled editorially. We look forward to future exciting submissions from your lab.

Sincerely,
